A realistic human head phantom for electromagnetic detection of brain diseases

Bai Zelin 1
Chen Diyou 2
Ma Ke 1
Jin Gui 3
Qiu Jinlong 4
Li Quanquan 1
Li Haocheng 5 lhc19890327@vip.qq.com
Chen Mingsheng 3 chenms83@tmmu.edu.cn
1 Material Procurement Room, Daping Hospital, Army Medical University , Chongqing , China
2 Department of Radiology, Daping Hospital, Army Medical University , Chongqing , China
3 College of Biomedical Engineering, Army Medical University , Chongqing , China
4 Institute for Traffic Medicine, Daping Hospital, Army Medical University , Chongqing , China
5 Department of Medical Engineering, General Hospital of Central Theater Command , Wuhan , China
Ravishankar Prashanth
Electronic publication date: 2025 Jan 29
Publication date: 2025
Volume: 13
Electronic Location ID: e18868
Received 2024 Aug 7; Accepted 2024 Dec 24
Copyright: © 2025 Bai et al.
Copyright year: 2025
Copyright holder: Bai et al.
License: This is an open access article distributed under the terms of the Creative Commons Attribution License, which permits unrestricted use, distribution, reproduction and adaptation in any medium and for any purpose provided that it is properly attributed. For attribution, the original author(s), title, publication source (PeerJ) and either DOI or URL of the article must be cited.
License URL: https://creativecommons.org/licenses/by/4.0/

Keywords: Dielectric properties, Electromagnetic simulation, Human head model, Stroke

Funding: National Natural Science Foundation of China 62171444 This research was supported by the National Natural Science Foundation of China under Grant 62171444. There was no additional external funding received for this study. The funders had no role in study design, data collection and analysis, decision to publish, or preparation of the manuscript.

==============================
The research on electromagnetic detection technology for brain diseases requires precise simulation of the human head. This article combines high-precision computed tomography (CT) images and magnetic resonance imaging (MRI) images to establish an electromagnetic numerical model of the human head with a real anatomical structure. (1) It had Asian characteristics and encompassed 14 different structures, including skin, muscles, cranial bones, cerebrospinal fluid, cerebral veins, cerebral arteries, gray matter, white matter of the brain, basal ganglia, thalamus, cerebellum, brainstem, eyeballs, and vertebrae. (2) The model used a combination of 0.625 mm-resolution CT and 1 mm-resolution MRI image data for reconstruction, with a smooth surface and high accuracy. (3) Within the simulation environment, this model enabled the generation of various brain disease scenarios, such as different types and degrees of cerebral hemorrhage and cerebral ischemia. It proved valuable for studying the distribution of electromagnetic fields in the human head and for investigating novel electromagnetic detection techniques exploiting brain tissue dielectric properties. (4) The created physical model and the numerical model were derived from the same person, which provided a good continuity between simulation experiments and physical experiments, and provided a realistic verification platform for the research of electromagnetic detection technology for brain diseases, such as differentiating the kind of stroke, monitoring brain edema, brain tumor microwave imaging, and diagnosis of Alzheimer’s disease.

Introduction

Electromagnetic detection technology is widely used by many scholars in the detection of brain diseases due to its advantages of speed, low cost, and easy integration and miniaturization (Scapaticci et al., 2018; Rodriguez-Duarte et al., 2021; Saied, Arslan & Chandran, 2022; Hossain et al., 2022, 2023; Ismail & Mustafa, 2023; Akazzim et al., 2024), such as stroke, brain tumor microwave imaging, and diagnosis of Alzheimer’s disease. The electromagnetic numerical model of the human head with real anatomical structure provides a realistic verification platform for the design and optimization of microwave detection technology for brain diseases before clinical experiments, and is an important tool for the research of microwave detection technology for brain diseases. The use of head models can simulate different types and degrees of brain diseases in simulation environments, and can quickly and low-cost obtain a large amount of experimental test data, providing rich training and testing sets for artificial intelligence based electromagnetic detection research of brain diseases (Tal & Shapira, 2019; Zhu et al., 2021; Hasan et al., 2023; Gong et al., 2023).

Most of the electromagnetic models of the human head used in current research are reconstructed from Caucasian image data, and models reconstructed from Asian image data are still rare (Alqadami et al., 2020; Abedi et al., 2021; Chen et al., 2022; Chen & Luk, 2022; Zhao et al., 2022). Currently, most of the head models used in electromagnetic detection research of brain diseases are integrated into electromagnetic simulation software, andmaking an independent realistic model is not trivial. Therefore, it is challenging to make realistic physical models for physical experiments based on numerical models used in simulation experiments. Due to this limitation, many physical and simulation experiments in research use different head models, greatly simplifying the structure of the head model. Some studies directly use the spherical model to replace the brain. Bisio et al. (2020), Fiser et al. (2022) use 2-D cylindrical models. Tobon Vasquez et al. (2020), Razzicchia et al. (2021) use 3-D single-tissue anthropomorphic ones. Those result in decreases of representatives for real scenarios, and the results of simulation experiments and physical experiments cannot be effectively verified with each other. Those limit their scope of application, and a more realistic model is desired for applying the research to clinical.

Meanwhile, several head models were created and adopted by microwave detection research for brain. Pokorny et al. (2019) designed and manufactured an realistic layered phantom of the human head from Caucasian anatomical image, but it was only simplified into five different layers that mimic the scalp, skull, cerebrospinal fluid, brain, and stroke regions. In Xiao, Tan & Dong (2019), used the “MIDA” model (Iacono et al., 2015) to investigate hemorrhage imaging by magnetic induction tomography. The “MIDA” model was originally established for analysis of safety and efficacy of medical devices with 1–2 mm resolution, which was modified and only five anatomical structures was involved in Xiao, Tan & Dong (2019). A reconstructed anatomical head model based on MRI scanning was used to investigate magnetic induction tomography to detect hemorrhagic stroke in Lv & Luo (2021), which comprised the cerebral cortex, skull, cerebrospinal fluid, gray matter, white matter, and the hematoma. The brain tissue composition was relatively rough. In Islam, Islam & Almutairi (2022), used metamaterial loaded compact directional 3D antenna to constructed a portable electromagnetic head imaging system and experimented with a model based on Caucasian image data. In the physical experiments measurements, another simple three-layer physical head model that was inconsistent with the numerical model was used. Origlia et al. (2023) presented the realization of a multi-tissue head phantom for testing microwave imaging devices. They reproduced unalterable components such as skin, fat, bone, grey matter, white matter, cerebellum, and ventricles using flexible and solid compounds made with proper proportions of urethane rubber, graphite powder, and salt to simplify multiple use. But it also limited the simulation of other types of brain diseases. More detailed and specific results are expected in future work. Therefore the development of a brain model with fine human brain structure for Asian people is of great significance to promote the development of microwave brain detection research.

This study combines high-resolution MRI and CT image data from a Chinese male healthy volunteer to reconstruct a human head electromagnetic numerical model with Asian characteristics for brain disease electromagnetic detection research. (1) It has Asian characteristics, which facilitates microwave brain detection researches for more than half of the world’s population; (2) It has characteristics of fine structures. It encompassed 14 different structures, including skin, muscles, cranial bones, cerebrospinal fluid, cerebral veins, cerebral arteries, gray matter, white matter of the brain, basal ganglia, thalamus, cerebellum, brainstem, eyeballs, and vertebrae; The model used a combination of 0.625 mm-resolution CT and 1 mm-resolution MRI image data for reconstruction, with a smooth surface and high accuracy. Where the cavity wall thickness in the physical model is only 1.5 mm. (3) The physical model is highly consistent with the numerical model. This numerical model can be applied to electromagnetic simulation software for simulation experiments, and can also be made into a physical model, so that the head models of simulation experiments and physical experiments come from the same model, enhancing the continuity and effectiveness of simulation experiments and physical experiments in the research of electromagnetic detection technology for brain diseases.

The establishment of this model is of great significance for the development and validation of electromagnetic detection methods for brain diseases. It enabled the production of different types and degrees of cerebral hemorrhage and cerebral ischemia, and had the potential to be applied to different brain disease scenarios. It was useful for studying the distribution of electromagnetic fields in the human head and for investigating novel electromagnetic detection techniques for brain diseases.

Electromagnetic numerical model

Acquisition of image data

Based on the imaging advantages of different image types, appropriate image data was used to reconstruct the corresponding structure. Table 1 provides the correlation between the structures to be reconstructed and the corresponding imaging data types. Both CT and MRI scans were performed using thin-layer scanning to ensure the accuracy of the reconstructed model. The CT and MRI scans had a layer resolution of 0.625 and 1 mm, respectively. The scanning was performed on a healthy 23-year-old Chinese male volunteer with no history of brain diseases or surgeries. The data were collected at the Army Characteristic Medical Center Daping Hospital. The study was approved by the ethics committee of the Chinese People’s Liberation Army Characteristic Medical Center (Medical Research and Ethics Review (2022) No. 147). The study obtained written informed consent from the study participants.

Table 1 Reconstruction structures and imaging types.

Tissue structure	Imaging type	
Bone	CTA arterial phase	
Muscle	CTA arterial phase	
Skin	CTA arterial phase	
Eyeball	CTA arterial phase	
Arteries	CTA arterial phase	
Veins	CTA venous phase	
Gray matter	MRI-T1WI	
White matter	MRI-T1WI	
Cerebellum	MRI-T1WI	
Brainstem	MRI-T1WI	
Thalamus	MRI-T1WI	
Basal ganglia	MRI-T1WI	
Cerebrospinal fluid	MRI-T2WI	

The CT scanning was performed using a 64-slice spiral CT scanner (64-slice LightSpeed; GE, Boston, MA, USA) at the Army Characteristic Medical Center Daping Hospital. The scanning range extended from the mandible to the cranial vertex. The specific scanning parameters included: a slice resolution of 0.625 mm, interslice spacing of 0.625 mm, tube voltage of 120 kV, tube current of 350 mA, and a pitch of 0.984. For the contrast-enhanced scan, an iodine contrast agent was injected through the elbow vein at a rate of 5 mL/s using a high-pressure injector, with a total volume of 50 mL. The arterial-phase scanning was conducted by the bolus tracking method, where the scan was triggered when the CT value in the region of interest (internal carotid artery) reached 150 Hounsfield units. The scanning direction was from the mandible to the cranial vertex. After completing the arterial-phase scanning, a 2 s interval was maintained, and then the venous-phase scanning was performed from the cranial vertex to the mandible. A total of 915 cross-sectional images were obtained for the arterial phase and 912 images were obtained for the venous phase during CTA scanning.

The MRI scans were conducted using a Siemens Verio 3.0 T superconducting MRI system with an eight-channel head coil. Two types of axial scans were performed: T1WI and T2WI. For the T1WI sequence, the imaging parameters were as follows: repetition time (TR) = 2,100 ms; echo time (TE) = 2.93 ms; matrix size = 256 × 256; slice thickness = 1 mm; and field of view (FOV) = 256 × 256 mm2. The total scanning time was 4 min and 16 s. A total of 160 cross-sectional images were acquired for the T1WI sequence. For the T2WI sequence, the imaging parameters were as follows: TR = 3,200 ms; TE = 409 ms; matrix size = 256 × 256; slice thickness = 1 mm; FOV = 250 × 250 mm2. The total scanning time was 9 min and 25 s. A total of 176 cross-sectional images were acquired for the T2WI sequence.

3D reconstruction of the model

This study used four different types of imaging data for 3D reconstruction: CTA venous phase, CTA arterial phase, MRI-T1WI, and MRI-T2WI. Initially, these four sets of imaging data were fused and registered (adjusting CT and MRI images to the same coordinate system to ensure that the structures reconstructed from different types of images can be restored to the head model). The CTA arterial phase data served as the template sequence and was registered with the other three types of imaging data as shown in Fig. 1. Following image fusion and registration, the fused and registered images were overlaid with the template CTA venous phase images for visual inspection. The largest error was identified and measured. If the maximum error was greater than 1 mm, the fusion and registration data were considered unacceptable. The process was repeated until the error was 1 mm or less. After successful registration, the corresponding tissue structures were segmented and reconstructed based on the image types described in Table 1. Both the image data fusion and tissue structure 3D reconstruction were carried out using the 3D Slicer software by an experienced technician proficient in human tissue structure 3D reconstruction. Figure 2 shows the segmentation and reconstruction of various organizational structures. After the segmentation of each tissue structure, the reconstructed segmented images were overlaid with the template sequence images within the designated region of interest. The largest error was identified and measured. If the maximum error exceeded 1 mm, the segmentation model was rejected and required reprocessing until the error was equal to or less than 1 mm. Once the segmentation models met the criteria, they were exported as STL format files.

Figure 1 Fusion registration of image data.

(A) Fusion registration of arterial and venous phases in CTA; (B) Fusion registration of CTA arterial phase and MRI-T2WI.

Figure 2 Segmentation and reconstruction of organizational structures.

(A) Reconstruction of skull; (B) reconstruction of arteries; (C) reconstruction of gray matter; (D) reconstruction of cerebellum; (E) reconstruction of veins; (F) reconstruction of eyeballs; (G) reconstruction of skin; (H) reconstruction of cerebrospinal fluid; (I) reconstruction of white matter; (J) reconstruction of brainstem; (K) reconstruction of basal ganglia; (L) reconstruction of thalamus.

Establishment of electromagnetic numerical model

In the CST electromagnetic simulation software, the 3D models of various tissue structures were imported in STP format using the “Import” function. Figure 3 depicts the human head model as imported into the CST software. Different cross-sectional views clearly displayed the various tissue structures inside the brain. The head model established in this study focuses on the influence of the dielectric properties of various parts of the head on the electromagnetic field. For tissues with similar dielectric constant and conductivity or smaller structures, they can be combined and simplified. For tissues which the dielectric constant and conductivity have a significant impact on the overall dielectric property of the head, they are reconstructed separately. The skin and skull have a large area and are the outermost tissues of the head, which is very important for studying the reflection of electromagnetic waves at the head boundary. In clinical practice, the main types of cerebral hemorrhage include basal ganglia hemorrhage, thalamic hemorrhage, cerebellar hemorrhage, and brainstem hemorrhage, with most cerebral hemorrhages occurring in the basal ganglia region. Therefore, the reconstruction of cerebral blood vessels, basal ganglia, brain, cerebellum, thalamus, and brainstem is crucial for electromagnetic detection research on cerebral hemorrhage. Cerebrospinal fluid is the most conductive part of the brain structure, so its reconstruction is also very important in the head model. Figure 4 illustrates the 14 different tissue structures that constituted the head model, namely skin, facial muscles, cranial bones, cerebrospinal fluid, gray matter of the brain, white matter of the brain, basal ganglia, thalamus, brainstem, cerebellum, cerebral arteries, cerebral veins, eyeballs, and vertebrae.

Figure 3 Electromagnetic model of the human head in the simulation software.

(A) a 3D view; (B) a sagittal plane view; (C) a transverse plane view; (D) a coronal plane view.

Figure 4 Structure of the tissues.

Assigning the appropriate dielectric parameters to each tissue structure at different frequencies created the human head electromagnetic numerical model. Considering the relationship between the size of the human brain and the wavelength of electromagnetic waves, the frequency of microwave brain detection is usually not more than 3 GHz. In this study, the electromagnetic model covered a frequency range of 0.3–3 GHz. Figure 5 presents the dielectric constants and tangential losses of blood, gray matter, white matter, cerebellum, cerebrospinal fluid, muscles, skin, nerves, vitreous humor (eye), and cancellous bone in the frequency range of 0.3–3 GHz, with data obtained from Gabriel’s database (Gabriel, Gabriel & Corthout, 1996; Gabriel, Lau & Gabriel, 1996). The dispersion spectrum of the tissue dielectric constants and tangential losses was sampled to generate 1,001 data points, facilitating the fitting of the material dielectric dispersion. Within the simulation software, a polynomial fit was used for dispersing each tissue material, with the fitting error controlled to be within 1%. The maximum fitting error for white and gray matters was 0.9734% and 0.1212%, respectively. After assigning the dielectric parameters to the different tissue materials, each material was designated for the corresponding structure in the human head model, resulting in the 0.3–3 GHz human head electromagnetic numerical model. The frequency range of the model could be adjusted based on specific needs by assigning the appropriate dielectric parameters to each tissue structure for the desired frequency point or range.

Figure 5 Dielectric constants and tangential losses of different tissue structures in the range of 0.3–3 GHz.

Stroke model

Various disease models, such as strokes, can be established based on a healthy head model. A brain hemorrhage electromagnetic numerical model was created by inserting a sphere at different positions within the brain and assigning the material of the sphere as blood. Figure 6 displays four different types of brain hemorrhage electromagnetic models: thalamic hemorrhage, brainstem hemorrhage, brain parenchymal hemorrhage, and basal ganglia hemorrhage. Different locations and degrees of brain hemorrhage could be simulated by changing the position and size of the blood sphere. Similarly, the sphere could be set with ischemic properties to establish a brain ischemia model, which implies having dielectric properties lower than those of the surrounding tissues by 10% (Semenov et al., 2017).

Figure 6 Electromagnetic simulation models for different types of brain hemorrhage.

(A) Brain stem hemorrhage model; (B) thalamic hemorrhage model; (C) basal ganglia hemorrhage model; (D) brain parenchymal hemorrhage model.

Electromagnetic simulation

The simulation software was CST Studio Suite 2020. The dielectric properties of each organization were derived from the database by Gabriel, Lau & Gabriel (1996). Within 0.3 to 3 GHz, 1,001 points were extracted from the database at equal intervals of frequency to fit the dielectric property dispersion spectrum of the brain tissue. In the simulation, the dispersion spectrum of brain dielectric properties was fitted by polynomial, and the fitting error was controlled within 1%. The maximum fitting error was 0.9734% for white matter and 0.1212% for gray matter.

Figure 7 demonstrates a brain hemorrhage simulation experiment conducted in the CST simulation software. A 1.2 GHz antenna was placed on the side of the head model for the simulation. The patch antenna used FR4 material as the substrate, and the metal copper foil was directly printed on the substrate. To achieve unidirectional radiation characteristics, the back side of the substrate, i.e., the ground layer, was completely covered with the metallic copper foil. The working frequency was around 1.2 GHz. A solid sphere was inserted into the right hemisphere of the brain. The material of the sphere was set to represent blood, simulating a brain hemorrhage. The radius of the sphere was set to 12.9 mm, representing a hemorrhage volume of 9 mL. Figure 7B shows the coronal section of the brain hemorrhage simulation experiment, the complete structure of the brain, as well as the location and size of the simulated brain hemorrhage. Figure 8 shows the S11 curves for 0 and 9 mL hemorrhage volumes. A significant difference was observed between the two curves near the operating frequency of the antenna. The Wilcoxon signed-rank test was performed on the S11 amplitudes for the two hemorrhage volumes within the frequency range of 1.2–1.216 GHz. The difference between the S11 amplitudes for the non-hemorrhage model and the 9mL hemorrhage model was statistically significant (P < 0.05), indicating that the two models had different impacts on the reflection coefficient of the antenna. Figure 9 presents the electric field and magnetic field distribution at a frequency of 1.2 GHz in the human head model. It was evident that using a high-precision human head model with a realistic anatomical structure was necessary for accurate simulation, enabling the realistic representation of actual scenarios. This approach allowed the observation and study of the electromagnetic effects of different brain tissues and the impact of the human head on electromagnetic wave propagation.

Figure 7 (A) Microwave patch antenna brain hemorrhage simulation experiment. (B) Cross-sectional view of the brain hemorrhage simulation experiment system.

Figure 8 S11 amplitude curves at 0- and 9mL bleeding volume.

Figure 9 (A) Human numerical head model electric field distribution at 1.2 GHz; (B) human numerical head model magnetic field distribution at 1.2 GHz.

About 9 h were cost in single-channel simulation (CST Studio Suite 2020 software, computer configuration: Intel(R) Core(TM) i7-10700K CPU @ 3.80 GHz, 32GB RAM, graphics card GTX1070Ti, hard disk with 1TB capacity), because the high-precision numerical model was created based on high-resolution MRI, CT, etc. For multi-channel microwave detection system, the number of channels is usually N(N-1)/2 (N is the number of antennas), which is generally time-consuming in simulation. However, the high-precision model is the foundation, and we can obtain a low-resolution model by down-sampling to adapt to different application scenarios. Secondly, it can also merge some small tissues to reduce the fineness of the brain structure and improve the simulation efficiency.

Construction of consistent physical model

Design and production of molds

Each brain tissue structure model was printed as a hollow model using 3D printing technology and then filled with tissue-mimicking materials. These individual components were then assembled to construct a physical human head electromagnetic model.

The design diagram of the 3D printing model is shown in Fig. 10A. The brain, cerebellum, brainstem, thalamus, and basal ganglia were all printed as hollow models that could be individually disassembled. The outer layer of the brain was a closed cavity filled with a liquid to simulate the cerebrospinal fluid. Similarly, the outermost layer of the head model was also a closed cavity filled with the appropriate material to simulate skin. All cavities that required filling with tissue-mimicking materials were perforated to ensure the efficient filling and drainage of these materials. The chosen 3D printing material was a fully transparent stereolithography resin (JS-UV-2015-T, the dielectric constant is about 3.6 ± 1) with precise and durable properties. Using a transparent material for printing facilitated the observation of the filling status of tissue-mimicking materials within the model. Figure 10B displays the 3D-printed transparent human head model, with each structure in a hollow form, and the cavities filled with appropriate tissue-mimicking materials. The corresponding plugs were also printed to seal the cavities. Of course, using this model data can also create other types of brain molds and develop corresponding brain physics models (Zhang et al., 2017; Mohammed et al., 2021; Särestöniemi et al., 2024).

Figure 10 (A) Design diagram for 3D printing of the model; (B) 3D printed human head model.

Production of physical models

The tissue-mimicking materials corresponding to the specific frequency were prepared and filled into different parts of the 3D-printed head model to create the human head electromagnetic model at the desired frequency. In this study, liquid tissue-mimicking materials for blood, gray matter of the brain, white matter of the brain, cerebrospinal fluid, cerebellum, skin, and nerves were prepared using pure water, 98% glycerol, anhydrous ethanol, methanol, and NaCl. The dielectric properties of these tissue-mimicking liquids were measured using a dielectric probe (Keysight N1501A) and a vector network analyzer (Keysight E5061B). The experimental setup for measuring the dielectric properties of tissue-mimicking liquids is shown in Fig. 11. Different liquids were added to a beaker and mixed thoroughly with a glass rod during the measurements. Care was taken to avoid air bubbles during the process. The measurements were performed once it was confirmed that no air bubbles were present in the beaker. To ensure accuracy, an adequate amount of liquid was used so that the bottom surface of the probe was at least 1 cm below the liquid surface. Table 2 lists the formulations of the tissue-mimicking liquids at 1.2 GHz. Table 3 presents the dielectric constants and tangential losses of the tissues and tissue-mimicking materials at 1.2 GHz. The dielectric properties of the tissues were obtained from Gabriel’s database. The tissue-mimicking materials exhibited a maximum deviation of 3.1% from the actual tissue dielectric constants (for nerves) and a minimum deviation of 0.26% (for skin). Regarding tangential losses, the maximum deviation from the actual tissues was 16.1% (for blood) and the minimum deviation was 0.59% (for gray matter).

Figure 11 Tissue-mimicking liquid dielectric characterization system.

Table 2 Formulations of tissue-mimicking liquids at 1.2 GHz.

Material	Blood	Brain gray matter	Brain white matter	Cerebrospinal fluid	Cerebellum	Skin (wet)	Nerve	
Water (mL)	5	4.5	4	5	4	4.5	3.5	
98% Glycerol (mL)	–	5.5	–	–	6	5.5	–	
Methanol (mL)	1.5	–	–	0.5	–	1	–	
Ethanol (mL)	–	–	6	–	–	–	6.5	
NaCl (g)	0.1	–	–	0.1	–	–	–	

Table 3 Dielectric properties of actual tissues and tissue-mimicking materials at 1.2 GHz.

Tissue name	Tissue	Tissue-mimicking material	
Dielectric constant	Tangent loss	Dielectric constant	Tangent loss	
Blood	60.561	0.41586	61.0175	0.4827	
Brain gray matter	51.565	0.31266	50.5408	0.3145	
Brain white matter	38.072	0.2705	37.7616	0.2413	
Cerebrospinal fluid	68.093	0.56126	67.4018	0.594	
Cerebellum	47.929	0.43783	46.274	0.4065	
Skin (wet)	45.109	0.31884	44.9933	0.3365	
Nerve	31.800	0.30850	30.8113	0.2934	

Different tissue simulation solutions were prepared according to the proportions in Table 2. The prepared tissue simulation solution was filled into the 3D printed model, and the filling holes were sealed with plugs and hot melt adhesive. The filled tissue structure was assembled to form a physical model of the human head at 1.2 GHz (a common frequency used in microwave brain detection studies (Persson et al., 2014; Gong et al., 2023; Abbosh et al., 2024)). Each structure was filled and installed according to the steps from inside out. First, the brainstem and cerebellar models were filled and installed. Next, the thalamus, basal ganglia, and brain models were filled and assembled. Finally, cerebrospinal fluid and skin simulation fluid were filled, as shown in Fig. 12.

Figure 12 Steps for constructing the human head electromagnetic model.

For ease of distinction and observation of different tissues, dye pigments were added to the tissue-mimicking liquids (Fig. 12). In actual experiments, dye pigments are not necessary to maintain the accurate dielectric properties of the tissue-mimicking liquids. Figure 13 presents an example where a balloon is inserted into the brain model. The blood-mimicking or ischemia-mimicking liquid can be injected into the balloon through a catheter, allowing the creation of actual electromagnetic models for cerebral hemorrhage and cerebral ischemia for physical experiments.

Figure 13 Electromagnetic model of stroke.

In the established physical head model, there were still some not significant difference with the numerical model. In order to accommodate the liquid that simulates the characteristics of brain tissue dielectric property, it was necessary to use 3D printing materials to make thin walls of different tissues, this was different from the numerical model, but it was essential in physical model. In order to minimize the impact of the wall of container, we made the container wall as thin as possible, the thickness was about 1.5 mm. We analyzed the effect of the wall of container in the model by simulation. In the experiment, the wall of container was endowed with the dielectric properties of the stereolithography resin. The results were shown in the Fig. 14. By comparing with the Fig. 9, it could be seen that the wall of container had some influence on the simulation results. Therefore, the ideal next step is to print containers with materials similar to the electrical properties of human brain tissue, although this is still relatively challenge.

Figure 14 (A) Physical head model electric field distribution at 1.2 GHz in simulation; (B) Physical head model magnetic field distribution at 1.2 GHz in simulation.

Electromagnetic measurement on the physical model

Based on the physical model made above (normal state, no brain disease), we conducted an electromagnetic measurement experiment, as shown in the Fig. 15. The antenna in the experiment was made according to Fig. 7 and connected to a vector network analyzer (Keysight 5061B). To avoid introducing interference, the antenna was placed on a specially 3D-printed shelf made of resin material. The measurement results of the experiment are shown in Fig. 15. By comparing the results in the Fig. 15 with the simulation results of the numerical model without intracranial hemorrhage in Fig. 8, it can be seen that the models could provide a good continuity between simulation experiments and physical experiments for human bain detection based on electromagnetic measurement.

Figure 15 Electromagnetic measurement experiment of physical model.

(A) Experimental setup of measurement; (B) result of measurement on the physical model.

Conclusion

This study reconstructed a 3D electromagnetic numerical model with authentic anatomical structures based on high-resolution CT and MRI thin-slice imaging data from a Chinese male individual. The high-resolution image data and reconstruction methods used in the reconstruction of this model ensure the authenticity and accuracy of the model. The model has good operability and visualization effects, enriching the human head electromagnetic model library based on Asian standards. This model allowed for the creation of different types of disease models, such as cerebral hemorrhage and ischemia, within a simulated environment. It could be effectively utilized to examine the electromagnetic field distribution in the human head and investigate new electromagnetic detection technologies relying on the electrical properties of brain tissues for diagnosing brain disorders. In addition, use the model data to create a brain mold and establish a physical model for real-world testing. Both the physical and numerical models in the software originated from the same human head model, ensuring a strong continuity between simulation and physical experiments in the research. This comprehensive model provided a realistic validation platform for assessing the effectiveness and safety of electromagnetic detection techniques for brain disorders, particularly for stroke detection, before proceeding to clinical trials.

Supplemental Information

Supplemental Information 1 Dielectric constant of organizational simulants.

Additional Information and Declarations

Competing Interests

The authors declare that they have no competing interests.

Author Contributions

Zelin Bai conceived and designed the experiments, performed the experiments, analyzed the data, prepared figures and/or tables, authored or reviewed drafts of the article, and approved the final draft.

Diyou Chen conceived and designed the experiments, performed the experiments, prepared figures and/or tables, authored or reviewed drafts of the article, and approved the final draft.

Ke Ma conceived and designed the experiments, prepared figures and/or tables, authored or reviewed drafts of the article, and approved the final draft.

Gui Jin performed the experiments, authored or reviewed drafts of the article, and approved the final draft.

Jinlong Qiu conceived and designed the experiments, prepared figures and/or tables, authored or reviewed drafts of the article, and approved the final draft.

Quanquan Li performed the experiments, prepared figures and/or tables, and approved the final draft.

Haocheng Li analyzed the data, authored or reviewed drafts of the article, and approved the final draft.

Mingsheng Chen analyzed the data, authored or reviewed drafts of the article, and approved the final draft.

Human Ethics

The following information was supplied relating to ethical approvals (i.e., approving body and any reference numbers):

The ethics committee of the Chinese People’s Liberation Army Characteristic Medical Center.

Data Availability

The following information was supplied regarding data availability:

The raw data are available in the Supplemental File.

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
