# Peer review of "A realistic human head phantom for electromagnetic detection of brain diseases"

_PeerJ, doi:10.7717/peerj.18868_

## Round 0.1 · original submission · Major Revisions

As the reviewers have mentioned, there are several clarifications that need to be made before being reviewed again. The manuscript holds potential to the journal's readership.

Reviewer 1 ·

Basic reporting

Some fragments are not clear. Please review the writing. See attached PDF.

Experimental design

I recommend extending the article's impact and scope and the state of the art. See the attached PDF for details and recommendations.

Validity of the findings

Improve the numerical experiment. See details on the PDF.

Additional comments

Bai, Chen, and colleagues present a numerical and physical head phantom that can be used to test EM-based techniques for the detection of brain diseases. They base the model on an Asian adult male and create it using CT and MRI images. The creation of the model and its release in a public database are important for the community on the field and are valuable contributions. However, some points can be improved.

Annotated reviews are not available for download in order to protect the identity of reviewers who chose to remain anonymous.

Reviewer 2 ·

Basic reporting

This article developes an electromagnetic simulation model of the human head with a real anatomical structure using CT and MRI images. Additionally, actual physical model is developed as well, by utilizing 3D simulation models in preparing realistic shaped molds for human tissue mimicking phantoms.
The topic is interesting and important since recently the importance of realistic models is understood. Additionally the developed models look very realistic.
However, there are several flaws which should be addressed:
1. Introduction and literature review is too narrow. There are also several realistic 3D head models presented in the literature which should be included in survey
2. Novelty is unclear and should be indicated clearly in the beginning: how this research and the developed model differentiates from the others presented in the literature
3. Discussion about computational complexity in simulation is such realistic models are used in the simulations e.g for stroke detection. In stroke detection, channel data has to be simulated between several antenna pairs around the head. If simulation model is so precise as in this case, the computational burden will be enormous (that’s why CST has voxel models since pixelized models are less complex despite of being anatomically realistic )
4. Human tissue mimicking phantoms: how the recipes are prepared, did the authors invent the recipe or is it from the literature (should be cited well if from the literature!)
5. Dielectric properties of transparent 204 stereolithography resin (JS-UV-2015-T) should be measured and presented. The impact of such material layer in the measurements should be discussed
6. Why this frequency range is chosen? Differences due to stroke seem to be very small… slightly higher frequency and better antenna would increase detectability of the stroke.
7. Since phantom model is presented, it would be important to present also measurement results with the similar antenna prototype as used in the simulations
8. Analysis of the results, including comparison between simulation and measurement results, are missing
9. English should be improved significantly
10. Paper stylizing should be improved (now there are different spaces between rows etc..)

Experimental design

Experimental design look very nice and realistic, but following issues should be addressed:
- what are the thickness and dielectric properties of stereolithography resin (JS-UV-2015-T) and how much they would affect on the results?
-S11 measurement results are missing with physical model. Should be compared with simulation results
-chosen frequency should be explaned
-methods should be explained better

Validity of the findings

-Measurement results with physical model are essential
-The simulation and measurements results should be analyzed, compared and discussed more in detail:
- How much stereolithography resin affects on the measurement results? (JS-UV-2015-T) and how much they would affect on the results?

-novelty should be emphasized

---

## Round 0.2 · Minor Revisions

As the reviewer's have stated, the revised submission has mostly addressed the original comments but there still needs to be some more clarification provided before the article is being reviewed.

Reviewer 1 ·

Basic reporting

Minor comments:
In 67, "it is impossible", it is challenging.
In 69, "Due to this limitation, many physical and simulation experiments in research use different head models." The actual reason is that making realistic phantoms is not trivial, not because the models are integrated with the software. This point gives more value to the work.
In 70, "directly use the spherical 71 model to replace the brain," they use simplified head models. Bisio 2020 and Fiser 2022 use 2-D cylindrical models. Tobon 2020 and Razzicchia 2021 use 3-D single-tissue anthropomorphic ones.
In 72, "effectiveness," the experiments are not less effective; they are less representative of a real scenario.
In 73, "the results of simulation experiments and physical experiments cannot be effectively verified 74 with each other." This is a strong statement. The mentioned references validated their simulated results with the experimental part. Again, the point is that simplified phantoms might not be representative enough of the actual head, limiting its scope. Then, for moving to clinical testing, it could be necessary to use more realistic phantoms, like the one proposed in the paper.

Experimental design

The methods for the 3D reconstruction of the model need some clarifications.
- Define image fusion and registration. What do you mean by fusion, and how is it done? Does the registration refer to taking either the MRI or the CT?
- Include more details in the segmentation. Is this done entirely manually, or is it used with another procedure, like U-net or another algorithm?
- Comment on the specific selections for the segmentation of each tissue (table 1).

Validity of the findings

Please check the color bar limits of Figure 14 for a better representation. For instance, looking at the figure, the blue (lowest limit) cannot be noticed. Moreover, considering the main interest is in the fields with the head, I recommend adjusting the maximum.

Reviewer 2 ·

Basic reporting

The authors have made several modifications on the paper based on the comments of the reviewers. Most of the comments were addressed well and the quality of the paper has been improved clearly. Literature review is now more comprehensive, though still bit narrow compared to the published studies in this field.
Professional structure is ok and results discussed well.

Experimental design

description on experimental design improved clearly

Validity of the findings

No new comments.

Additional comments

The description of frequency selection is still unclear. Most of the stroke detection studies use either ISM bands or lower UWB band. Actually I have not seen other stroke detcetion studies at 1.2 GHz.

---

## Round 0.3 · accepted · Accept

Thank you for addressing all the reviewer comments to satisfaction.

Reviewer 1 ·

Basic reporting

no comment

Experimental design

no comment

Validity of the findings

no comment

Additional comments

Thanks to the authors for handling my comments and questions. The article has improved from its initial form and is now acceptable for publication.

Reviewer 2 ·

Basic reporting

The authors have made the required modifications now and the paper is ready to be accepted from my side.

Experimental design

The authors have made the required modifications now for this study part and the paper is ready to be accepted from my side.

Validity of the findings

The authors have made the required modifications now for this study part and the paper is ready to be accepted from my side.

Additional comments

The authors have made all the required modifications now and the paper is ready to be accepted from my side.